# Competitive Relationship between Flood Control and Power Generation with Flood Season Division: A Case Study in Downstream Jinsha River Cascade Reservoirs

**Hongyi Yao [1], Zengchuan Dong [1],\*, Wenhao Jia [1], Xiaokuan Ni [1], Mufeng Chen [1], Cailin Zhu [2] and Dayong Li [1]**

1    College of Hydrology and Water Resource, Hohai University, Nanjing 210098, China;
     yaohy@hhu.edu.cn (H.Y.); wenhao@hhu.edu.cn (W.J.); hynxk@hhu.edu.cn (X.N.);
     chenmf@hhu.edu.cn (M.C.); lidayong@hhu.edu.cn (D.L.)
2    Shanghai Hydrology Administration, Shanghai 20032, China; zhucailin_hhu@163.com
\*    Correspondence: zcdong@hhu.edu.cn

**Abstract:** The lower reaches of Jinsha River host the richest hydropower energy sources in China. With the construction of Wudongde and Baihetan, the multi-objective optimization for cascade reservoirs (along with Xiluodu and Xiangjia Dam) in the lower reaches of Jinsha River will create significant benefits. This paper focuses on the competitive relationship between flood control and power generation, and attaches attention to the measurement of different objective functions and their competitive relationship. With observations of the flood in 1974, 1981, and 1985, a 100-year return period flood with peak-3d volume pair as different inputs for the optimal model is approached by NSGA-II. Different flood seasons divided by flood feature is applied to figure out specific competitive relationship. The results can be concluded as the following: (1) Strong competitive relationship mainly occurs in pre-flood season. (2) Whether it shows a strong competitive relationship depends on the amount of discharge. If the turbine is set to full capacity, power generation is fulfilled certainly, which means that there exists a weak competitive relationship between multi-objectives. (3) The different processes of floods have an effect on the duration of a competitive relationship. A flood with a late peak causes the extension of strong competition in the pre-flood season, which lends itself to a strong competition relationship in the post-flood season. (4) The intensity of competition in the pre-flood season is higher than that in the post-flood season because it has a larger range.

**Keywords:** reservoir optimal operation; flood control and power generation; competitive relationship; copula; NSGA-II; flood season division

## 1. Introduction

Flood disaster is one of the most destructive natural disasters [1]. In order to mitigate flood disasters, people build water conservancy projects such as reservoirs to reduce flood risks. Reservoir operation is an important means of flood control and disaster reduction, which has been studied by many scholars in the past decades [2–4]. However, with the development of hydrological forecasting, meteorological forecasting, and computer technology, the ability of utilizing reservoir operations to reduce flood damage improves. Many investigators propose some other purposes like hydropower generation, water supply, ecology, and irrigation during flood season [5,6]. All of these objectives, along with flood control, constitute the multi-objective optimal operation of reservoir systems in the flood season. Taking the multi-objective optimal operation of flood period into account, we can make

efficient use of the large amount of water in the flood season and make considerable benefits, while ensuring flood control safety. Braga et al. [7] applied a combination of simulation and optimization to realize the tradeoffs between flood control and hydropower generation. Hojjati et al. [8] established a multi-objective optimal dispatch model for reservoir systems with maximum generation capacity and flood storage volume. Ngo et al. [9] applied a combination of simulation and optimization to realize the tradeoffs between flood control and hydropower generation, to guide the operation of reservoirs. Liu et al. [10] reflected the four objectives of upstream flood control, downstream flood control, power generation, and navigation. They pointed out that the Three Gorges Reservoir could increase the benefits of power generation and navigation under the premise of ensuring flood control safety in the case of small and medium floods. Zhou Jianzhong et al. [11] proposed the balanced flood control index and applied it in a comprehensive utilization model in the flood season to ensure the tasks of flood control, navigation, and power generation. They improved the existing dispatching schemes and formulated dispatching schemes according to different priorities. Xu Bin et al. [12] studied the three objectives of flood control, power generation, and water supply of cascade reservoir system, and analyzed the variation of power generation benefits under different initial water levels of the Three Gorges Reservoir.

Although multi-objective optimization has been widely discussed in reservoir dispatching systems, and some fruits have been applied in practice, the competitive relationship between multi-objectives has been given limited attention and its rules and variations are unknown. Most of the existing studies start from the Pareto front to distinguish the competitive relationship between the two objectives in the whole research period [13]. Some scholars think that taking several objectives into account is to reduce the competition between objectives [9,14]. Raupp et al. [15] proposed a new method for calculating "waiting volumes", which can minimize the conflict of flood control and hydroelectric generation. However, its weakness lies in that these two objectives in competition are the former in rainy season and the latter in dry season. Li [16] thought that the contradiction between the minimum flood loss and the maximum storage level at the end of the period depended on the initial water level of the reservoir, but did not give a detailed explanation. Meng [17] proposed that for the two objectives of power generation and water supply, the relationship between the two is not always competitive during the research period, and he found out that the competition relationship decreases with probability of water supply increases. Huang [18] pointed out that asymmetry trade-offs existed among power generation, water supply, and ecological objectives in reservoir operation.

For the study on flood season division, people pay more attention to the study of dynamic control of flood limited water level to improve the efficiency of water use [19,20], rather than the change of competitive relationship between research objectives. In general, there is little discrimination in current research on the internal competitive relationship between the two objectives in reservoir flood control dispatching, nor the relevant research on the strength of the competitive relationship. Therefore, this paper studies mapping relationships between the integrated varying trend of different objective function values and the competitive relationship. It divides research period in sections and then calculates competitive relationships between the objectives. The competitive strength coefficient is quantitatively calculated, and the rationality of the competitive strength is analyzed in combination with the operation state of the reservoir.

## 2. Study Area

The Jinsha River is located in the upper reaches of the Yangtze River and its main stream is 3481 km long. It flows through four provinces (districts), including Qinghai, Tibet, Sichuan, and Yunnan. The slopes are steep and the flow is abundant and stable. The water energy reserves are about 113 million kW, which accounts for 42% of the hydropower resources in the Yangtze River Basin. It is a "rich mine" for the development of China's hydropower resources and an important energy base for the "West-to-East Power Transmission" strategy [21,22]. At present, the two reservoirs of Xiluodu and Xiangjiaba in the lower reaches of the Jinsha River have been put into use, while the Wudongde and

Baihetan reservoirs are under construction. The four cascade reservoir systems jointly undertake a series of tasks such as flood control, power generation, and navigation in the basin. A map of the considered system is shown in Figure 1. The flood season of downstream Jinsha River is from 1 July to 10 September, and the relevant reservoir parameters are shown in Table 1. This paper conducts research on flood control and power generation optimization of the four reservoirs in the lower reaches of the Jinsha River. In this paper, the competitive relationship between the two objectives of flood control and power generation is calculated to guide the actual operation of the cascade reservoirs.

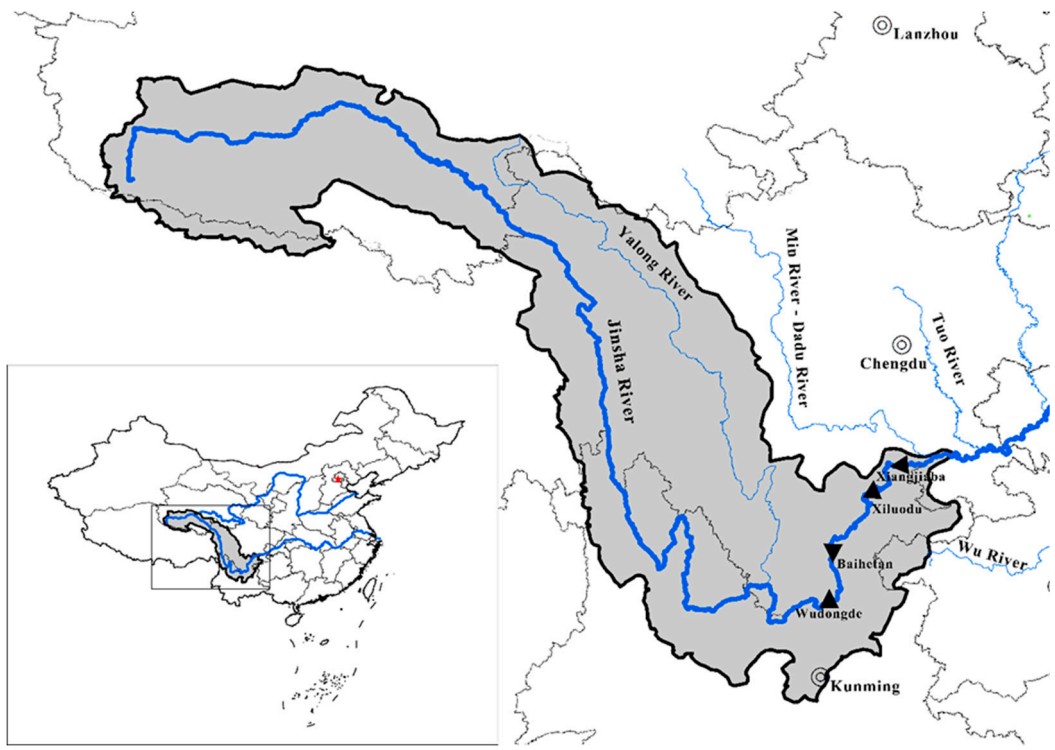

**Figure 1.** Location sketch map of the downstream Jinsha River cascade reservoir systems.

**Table 1.** Reservoir parameters during flood season.

| Reservoir | Lower Bound of Water Level (m) | Upper Bound of Water Level (m) | Lower Bound of Discharge (m³/s) | Upper Bound of Discharge (m³/s) | Installed Capacity (MW) | Maximum Allowable Water Discharge Variation Rate (m³/(s·d)) |
|---|---|---|---|---|---|---|
| Wudongde | 952 | 972 | 883 | 30,000 | 10,200 | 2000 |
| Baihetan | 785 | 825 | 700 | 30,000 | 16,000 | 2000 |
| Xiluodu | 560 | 600 | 1060 | 30,000 | 13,860 | 2000 |
| Xiangjiaba | 370 | 380 | 830 | 30,000 | 6000 | 2000 |

## 3. The Joint Optimal Operation Model of Reservoir Model

### 3.1. Model Construction

#### 3.1.1. Objective Function

In the joint optimal operation model of cascade reservoir systems, the minimal downstream flooding risk of the flood control objective is determined by the minimum sum of squares of the discharge flow [23,24] of the downstream reservoir, which is almost equivalent to the minimum of the largest discharge of the dam:

$$F_1 = min \sum_{t=1}^{T} q_{last,t}^2 \tag{1}$$

where $q_{last,t}$ is the discharge volume of the last reservoir of the cascade reservoir system during $t$-th period (m³/s); $T$ is the number of periods (d).

Maximizing power generation of the reservoir system during the period:

$$F_2 = max \sum_{t=1}^{T} \sum_{i=1}^{n} k_i q_{i,t} H_{i,t} \Delta t \tag{2}$$

where $k_i$ is the output coefficient of the $i$-th reservoir; $q_{i,t}$ is the discharge volume of the $i$-th reservoir during $t$-th period; $H_{i,t}$ is average water head of the $i$-th reservoir during the $t$-th period (m); $\Delta t$ is the duration of each period (h); $n$ is the number of reservoirs in the cascade reservoir system; $T$ is the total number of days in the flood season (d).

### 3.1.2. Constraints

(1) Water balance constraint

$$V_{i,t+1} = V_{i,t} + (Q_{i,t} - q_{i,t}) \times \Delta t \ i = 1 \tag{3}$$

$$V_{i,t+1} = V_{i,t} + (Q_{ri,t} + q_{i-1,t} - q_{i,t}) \times \Delta t \ i = 2, 3, \ldots n \tag{4}$$

where $V_{i,t}$ is the reservoir storage volume of the $i$-th reservoir during $t$-th period (m³); $Q_{i,t}$ is the inflow of the $i$-th reservoir during $t$-th period (m³/s); $Q_{ri,t}$ is the local inflow of the $i$-th reservoir during $t$-th period (m³/s).

(2) Water level constraint

$$Z_{i,t}^{min} \leq Z_{i,t} \leq Z_{i,t}^{max} \tag{5}$$

where $Z_{i,t}$ is water level at the end of the $t$-th period (m), $Z_{i,t}^{min}$ and $Z_{i,t}^{max}$ are upper and lower bounds, respectively, at the end of the $t$-th period (m).

(3) Discharge volume constraint

$$q_{i,t}^{min} \leq q_{i,t} \leq q_{i,t}^{max} \tag{6}$$

where $q_{i,t}$ is the average discharge volume of the $i$-th reservoir during the $t$-th period (m³/s); $q_{i,t}^{min}$ and $q_{i,t}^{max}$ are the lower limit and the upper limit of the discharge volume of the $i$-th reservoir during the $t$-th period, respectively (m³/s)

(4) Discharge variation constraint

$$\left| q_{i,t} - q_{i,t+1} \right| \leq \Delta q \tag{7}$$

where $q_{i,t}$ is the average discharge volume of the $i$-th reservoir during the $t$-th period (m³/s); $q_{i,t+1}$ is the discharge volume of the $i$-th reservoir during the $t$+1-th period (m³/s); $\Delta q$ is the upper limit of the discharge variation (m³/s).

(5) Installed capacity constraint

$$N_{i,t} \leq N_{i,C} \tag{8}$$

$$N_{i,t} = k q_{i,t} H_{i,t} \tag{9}$$

where $N_{i,t}$ is the output of the $i$-th reservoir during the $t$-th period (kW); $N_{i,C}$ is the installed capacity of the $i$-th reservoir (kW); $k$ is the efficiency coefficient of the turbine; $q_{i,t}$ is the discharge volume of the $i$-th reservoir during the $t$-th period(m³/s); and $H_{i,t}$ is the average water head of the $i$-th reservoir during the $t$-th period (m).

*3.2. Input*

### 3.2.1. Measured Flood

According to historical records, large floods occurred in the vast areas in the middle and upper reaches of the Yangtze River in 1974, 1981, and 1985, and their recurrence periods are about 90, 20, and 25 years, respectively, causing huge losses. However, the time and high flow duration of the flood peaks are different, so it is of great significance to select the measured runoff in the three flood seasons as model inputs. The reservoir inflow of the Jinsha River in the three years is shown in Figure 2.

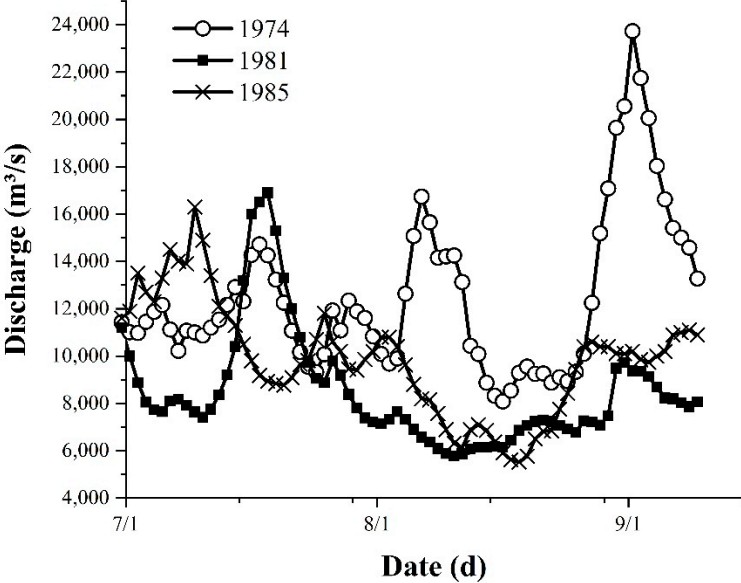

**Figure 2.** Inputs of measured flood discharge trajectory.

### 3.2.2. Design Flood

In order to further explore the influence of flood processes of different magnitudes on the competition of objective functions, this paper uses Copula function to amplify typical floods. Copula originally meant "joining", which can connect multiple arbitrary forms of margins to form a multidimensional joint probability distribution function [25]. Therefore, the Copula function can be used as a new idea to construct a joint distribution of flood peak flow and multi-day flood volume to estimate the design flood hydrograph more precisely [26]. Specific steps are as follows:

(1) Select the flow data of 16 years from 1971 to 1986 forHuatan, Ningnan, Xiaohe, Meigu, Xining, and Xinhua sites (Figure 3 shows site distribution). Since Wudongde did not establish a site at that time, the measured flow rate by the Huatansite is used as the inflow. The inflow of Xiluodu and Xiangjiaba ignored the flood evolution, which means the input is added by Xiaohe and Meigu, and Xining and Xinhua, respectively. Then, calculate the marginal distribution of flood peak flow and flood volume in three days.

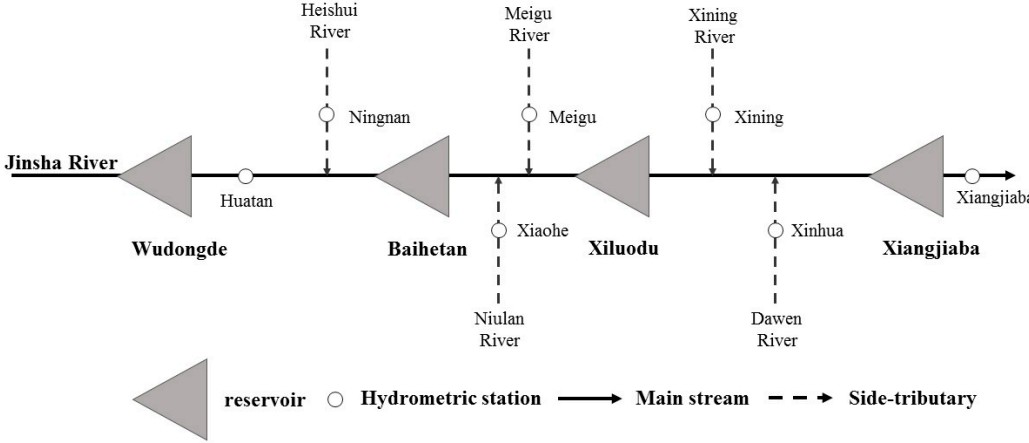

**Figure 3.** The schematic diagram of Jinsha River downstream cascade reservoirs.

(2) The binary joint distribution of flood peak flow and 3d (3-day) volumes are constructed by Clayton Copula, Frank Copula, and Gumbel Copula, respectively.

(3) Use the Monte Carlo simulation to compare the simulated values with the measured values. Take the Huatan site as an example, and the results are shown in Figure 4.

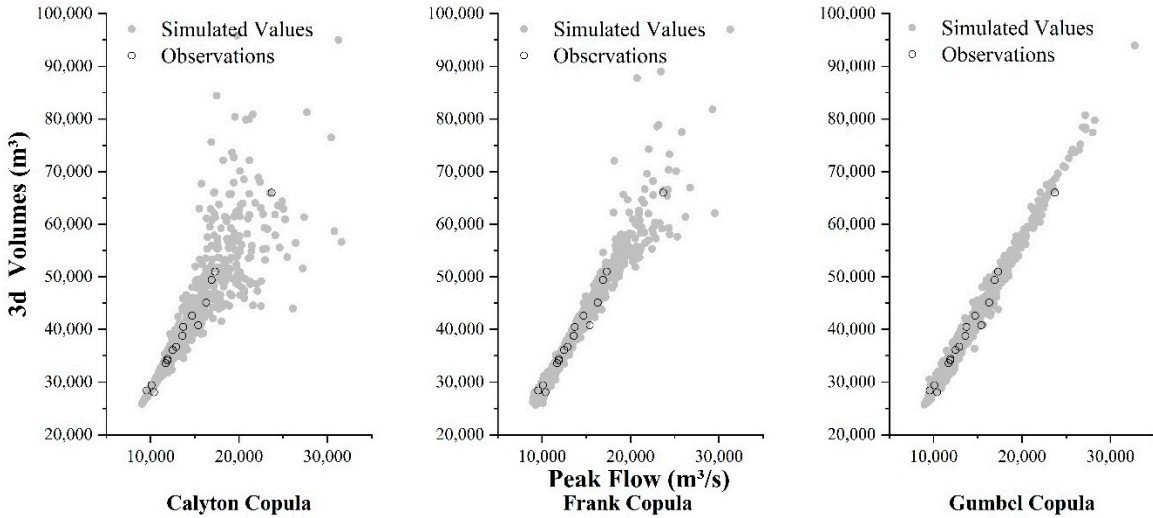

**Figure 4.** Observations versus simulations for the three types of Copulas at Huatan Station.

(4) Taking the root-mean-square error (RMSE) as an indicator, the fitting performance can be calculated [27], and the Copula function with the minimum RMSE (in bold) is selected for each inflow condition. The calculation results are shown in Table 2.

**Table 2.** Root-mean-square error (RMSE) for each node.

| Name | Clayton Copula | Frank Copula | Gumbel Copula |
|---|---|---|---|
| **Inflow of Wudongde** | 0.0620 | 0.0589 | **0.0579** |
| **Interval inflow of Baihetan** | 0.0825 | 0.0759 | **0.0753** |
| **Interval inflow of Xiluodu** | 0.0735 | 0.0678 | **0.0676** |
| **Interval inflow of Xiangjiaba** | 0.0848 | **0.0686** | 0.0723 |

(5) According to the selected Copula function, the co-occurrence and return period for 100 years of peak flow and flood volume in three days are construed and shown in Table 3.

**Table 3.** Simulated values of peak flow and 3d volumes.

| Name | Co-occurrence and Return Period = 100 y | |
| --- | --- | --- |
| | Peak Flow (m³/s) | 3D volumes (m³/s·d) |
| Inflow of Wudongde | 26,000 | 73,700 |
| Interval inflow of Baihetan | 763 | 1530 |
| Interval inflow of Xiluodu | 1540 | 4550 |
| Interval inflow of Xiangjiaba | 797 | 1030 |

(6) The new amplification method of the typical flood process proposed by Xi et al. [28] is applied to maintain the characteristics of typical floods:

$$
\begin{aligned}
\text{minf} = \sum_{i=2}^{72} \left| \frac{Q_D(i)-Q_D(i-1)}{\Delta t} - \frac{Q_M(i)-Q_M(i-1)}{\Delta t} \right| + M_1 \times \left| Q_D^{max} - Q_p \right| \\
+ M_2 \times \left| \sum_{j=start}^{end} Q_D(j) \times \Delta t - W_3 \right|
\end{aligned}
\tag{10}
$$

where $Q_D(i)$ and $Q_M(i)$ are design flood and measured flood, respectively; $i$ is the sequence number of date; $Q_p^{max}$ is the maximum discharge of design flood; and $Q_p$ is the simulated peak flow. $\sum_{j=start}^{end} Q_D(j)$ is the maximum three-day volumes of design flood and $W_3$ is the simulated 3d volumes. $M_1$ and $M_2$ are coefficient of penalty function which are large enough. The optimization problem is solved by POA, and then we get the design flood trajectory.

The traditional amplification method in China only focuses on the peak flow, leading to an unfixed and unsmoothed trajectory of design flood. This new amplification method can satisfy the dual-restriction (peak flow and 3d volumes) and produce the only design flood from measured flood without manual correction. Figure 5 shows two scenarios of flood (take the year 1985 and inflow of Wudongde for example) to confirm that the design flood keeps the mode of measured flood and the dual-restriction are met.

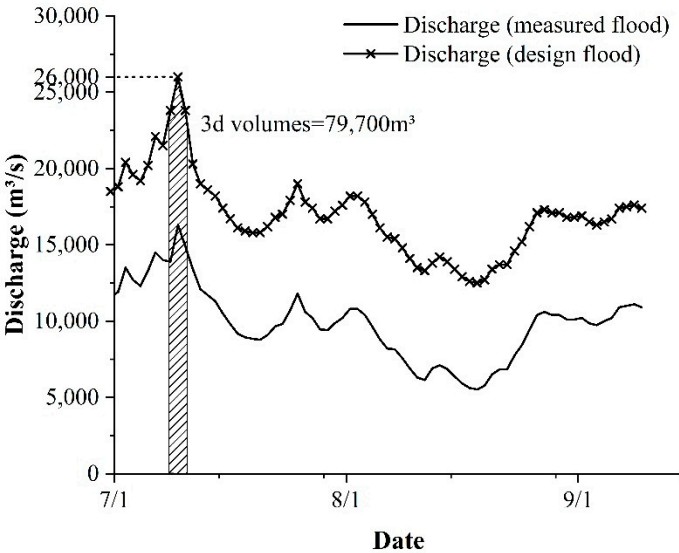

**Figure 5.** Comparison with measured flood and design flood.

*3.3. Model Solution*

NSGA-II [29] is an efficient multi-objective optimization algorithm and has a wide range of applications in reservoir dispatch [8,30]. In this paper, the algorithm of NSGA-II is selected and solved based on the multi-objective optimization platform developed by Tian [31]. The flow chart of NSGA-II can be seen in Figure 6. Specifically, the constraints (1) to (4) are met in section "Repair infeasible

solutions" and the installed capacity constraint is considered in fitness function. Variables ($Z_{i,t}$ and $q_{i,t}$) that have an excess or deficiency of limits should be adjusted into limits, while the excess output does not count. According to the relevant planning, the flood season in the lower reaches of the Jinsha River is from 1 July to 10 September. As the decision variable, discharge flow involves a total of $72 \times 4 = 288$. The population size is 120 and the number of evaluation is 120,000. The crossover operator is selected as simulated binary crossover (SBX) and the mutation operator is selected as polynomial mutation (PM). Three types of measured flood processes and three types of amplified flooding process in the 1974, 1981, and 1985 floods (referred to as inflows 1-1, 1-2, 2-1, 2-2, 3-1, and 3-2 hereafter) are chosen as the inputs, attempting to solve the flood control-power generation optimal operation model.

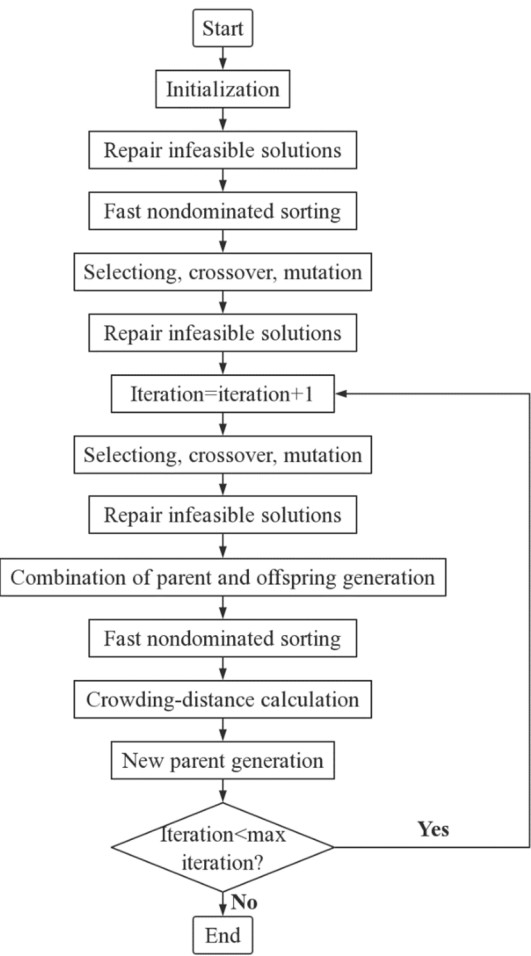

**Figure 6.** Flow chart of the NSGA-II algorithm.

## 3.4. Discrimination of Competitive Relations

It is difficult to optimize multiple objectives simultaneously. Unlike single-objective optimization problems, multi-objective problems do not get a solution, but a set of solutions consisting of many nondominated solutions, called Pareto solution sets [32]. It is believed that if the Pareto solution set can converge to a smooth curve, then the objectives are in competitive relationship. However, the Pareto frontier often represents the total result of the whole study period, which means it cannot analyze the competitive relationship of the internal objectives within the sub-period during the study period. Thus, a new competitive relationship discriminating method is needed. Selecting the shortest point between the Pareto optimal set and the mean Euclidean distance, the new discriminant method is established by finding mapping relationship between variation of the objective function and competitive relationship of the objectives.

Because there is a hydraulic connection between the upstream and downstream of the reservoir, it is assumed that the discharge volume of the last reservoir (Xiangjiaba) in the decision variable ($q_{last}$) is an independent variable. In this paper, two objectives are $F_1 = min \sum f_1$ and $F_2 = max \sum f_2$. When $f_1'(x) \times f_2'(x) > 0$ in a certain sub-period, two objective functions increase or decrease simultaneously, which confronted with the overall objective, so the two objectives show strong competition. On the contrary, when $f_1'(x) \times f_2'(x) \leq 0$, there is no competition between the two objectives. Table 4 establishes the mapping relationship between the varying trend of the objective functions $f_1$ and $f_2$ and their degree of competition. As for the trend testing approaches, Mann–Kendall's (M–K) non-parametric test method is applied [33,34]. The specific process is as follows:

Find the shortest point in the Pareto solution set to the mean Euclidean distance as the most representative scheduling solution.

The M–K test is used to examine the segmentation trend test on the two objective function process lines of the selected Pareto solution set.

The competitive relationship of divided–period, as shown in Table 4, is obtained as a representative competitive relationship between objectives.

To further determine the strength and weakness of the strong competitive relationship, the expression of the competitive strength coefficient $\eta$ is derived by:

$$\eta = \frac{\partial E}{\partial q_4} \cdot \frac{\partial q_4^2}{\partial q_4} = \frac{\partial E}{\partial t} \cdot \frac{\partial t}{\partial q_4} \cdot \frac{\partial q_4^2}{\partial t} \cdot \frac{\partial t}{\partial q_4} \tag{11}$$

In the formula, $E = N_{sum} \cdot \Delta t$ means the total power generation of reservoir system in the $\Delta t$ period. $N_{sum}$ is the total output of the reservoir system during the $\Delta t$ period. The rest of the symbols have already been mentioned before. The differential equation is converted to difference equation:

$$\eta = \frac{\Delta E}{\Delta t} \cdot \frac{\Delta q_4^2}{\Delta t} \cdot \left(\frac{\Delta t}{\Delta q_4}\right)^2 + \delta \tag{12}$$

where $\delta$ is the sum of the error terms resulting from the differential conversion to the difference, which is ignored in subsequent calculations.

**Table 4.** Competitive relationship with maxF1 and minF2.

| Competitive Relationship — Obj1 / Obj2 | Upward Trend | No Apparent Trend | Downward Trend |
|---|---|---|---|
| **Upward trend** | strong | weak | no |
| **No apparent trend** | weak | weak | weak |
| **Downward trend** | no | weak | strong |

## 4. Results

According to the obtained objective function value, the Pareto front is drawn as shown in Figure 7.

The results show that flood control and power generation present an interactional and mutually contradictory relationship. It is clear that maximum power generation and minimum flood damage cannot be achieved simultaneously. The risk of flooding increases with the increase in power generation. With the increase of incoming water, the increase of power generation in 1-1 and 1-2 is not obvious with the increased average value only by 0.5%. But the power generation in Series 2 and Series 3 increases significantly, with a growth rate of 11.9% and 1.8%, respectively. Another objective, flood control risk, increases significantly, with increases of 46.7%, 231%, and 248%, respectively. In order to

further explore the rules of variation in the competitive relationship and their internal mechanisms, it is necessary to divide the time period to discriminate the competitive relationship.

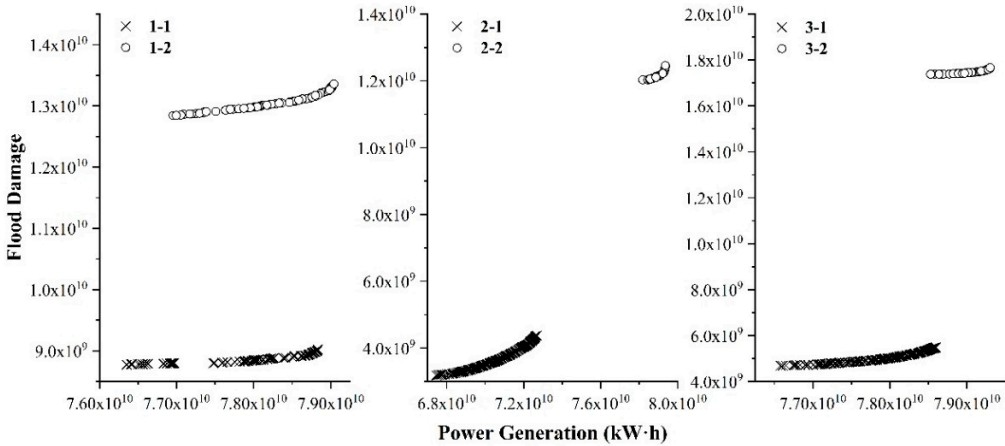

**Figure 7.** Multi-objective Pareto curve for flood control and power generation with different inflow.

This paper divides time period based on the fluctuation of the flood process and the important nodes of the reservoir operation and then judges the trend of the flood control and power generation through the hydrographs. The result of their competition is shown in Figure 8. Overall, there is a strong competitive relationship between the two objectives in the early period of the flood season, then it weakens in the medium term and increases at the end of the period. However, in the scenarios of 2-2 and 3-2, the weak competitive relationship continues to be maintained to the end. For inflows 1-1 and 1-2, the flood characteristics are different from other scenarios of incoming water; the flood volume at the end of the period is very large and the flood peak comes late. Meanwhile, the strong competition in the early period lasts for 40 days, which is much longer than other incoming water scenarios. After a short period of about 15 days of weak competition, it becomes a strong competition with the arrival of flood peak.

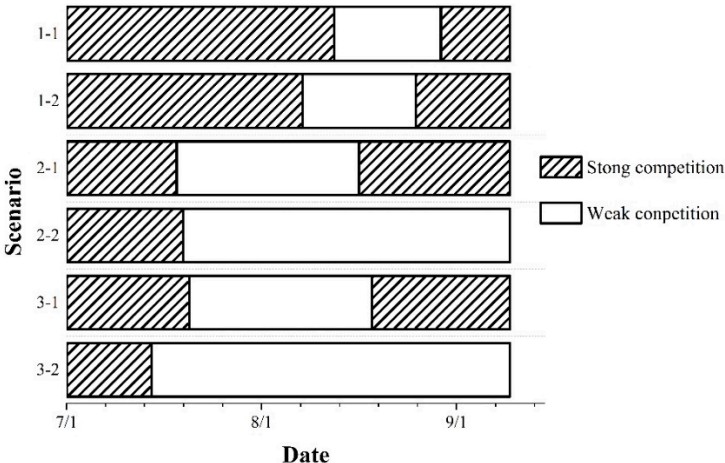

**Figure 8.** Competition results of power generation and flood control.

In order to quantitatively describe the strength of the strong competitive relationship, the competitive strength coefficient $\eta$ is calculated according to Equation (12). The results are shown in Table 5. In the table, $\eta 1$ and $\eta 2$ are the competitive strength coefficients of the pre-flood season and the post-flood season, respectively. In the same dispatching process, the competition coefficient at the beginning of the flood season is obviously larger than that late in the flood season, which means that in strong competition relationship, the competition in the early period of the flood season is more

intense, which should become the key research period of dispatching decision-making. Compared with the same pattern, along with different magnitudes of the flowing process, the competition intensity coefficient of the amplified flood in the early stage is weaker, which is 37.8%, 20%, and 41.8% lower than that of the typical flood.

**Table 5.** Chart of coefficient of competitive intensity.

| Scenario | $\eta 1$ | $\eta 2$ |
|:---:|:---:|:---:|
| 1-1 | $8.2 \times 10^9$ | $6.6 \times 10^7$ |
| 1-2 | $5.1 \times 10^9$ | $7.3 \times 10^7$ |
| 2-1 | $6.5 \times 10^9$ | $2.0 \times 10^9$ |
| 2-2 | $5.2 \times 10^9$ | $--$ |
| 3-1 | $1.1 \times 10^9$ | $9.0 \times 10^7$ |
| 3-2 | $6.4 \times 10^8$ | $--$ |

## 5. Discussion

In order to further explore the causes of the competitive relationship, the measured flow process (3-1) and its amplified flow process (3-2) in 1985 were selected as an example for analysis. The characteristic parameters of each reservoir are shown in Figure 9. For the objective of power generation, the way of "high water head and large flow" is the ideal situation to maximize the power generation. However, for the objective of flood control, the "planarization" of discharge process can effectively reduce the downstream flood control risk. Therefore, the competition between the two objectives is to satisfy the distribution of water under the constraint conditions, and this competition is particularly obvious when the power generation is very small in the early stage of the flood season. It can be seen from the figure that in the early period of the flood season, the four reservoirs choose to rapidly raise the water level in order to ensure that the turbine unit reaches the full state as soon as possible, and at the same time, to provide protection of high water head for future power generation. The flood peak appears around 10 July, so it is necessary to release water in time according to the flood volume so as to avoid damage to the dam. Thus, there is a strong competitive relationship between the two objectives at this stage. After the middle of the flood season, the inflow rate is significantly reduced, the flood control pressure is also reduced, and the reservoir can meet the flood control requirements only by properly releasing the water. At this time, the objective of power generation can be achieved easily with the high water head accumulated before. Thus, the two objectives of flood control and power generation are in weak competition. At the end of the flood season, there are two different situations. In scenario 3-1, before the arrival of the small flood peak at the end of the period, the water levels of the two reservoirs in the most upstream Wudongde and Baihetan decline, resulting in a decrease in power generation. Then, the competition becomes intense in subsequent power generation as the water level rises again. This stage can be seen as the reproduction of strong competition in the first stage. However, the flow volume is much smaller than the first stage, and variable range of water level is less than that in the first stage, so the competition intensity is also smaller than that in the first stage. In scenario 3-2, although the water levels of Wudongde and Baihetan also decrease, the turbine unit of reservoir can continue to exert its full force and the power generation remains almost unchanged, so the two objectives are still in weak competition.

Scenarios 2 and 3 have similar competition results because of similar reservoir inflow flood. In scenario 1, the competition results are slightly different due to different flood characteristics. It can be seen from the output hydrograph of Figure 10. In the early stage, the incoming water is small, and the power generation rises but it does not reach full capacity until the 40th day. So, the strong competition relationship lasts for 40 days. After a period of full force and weak competition, the reservoir system needs to prepare for flood peak at the end of the period, so the strong competition relationship reappears, which is similar to that of scenarios 1 and 2. At this time, subject to water level of the

reservoir, the arrival of flood peak intensifies the competition between power generation and flood control, making the competition more intense during this period greater than that of the beginning.

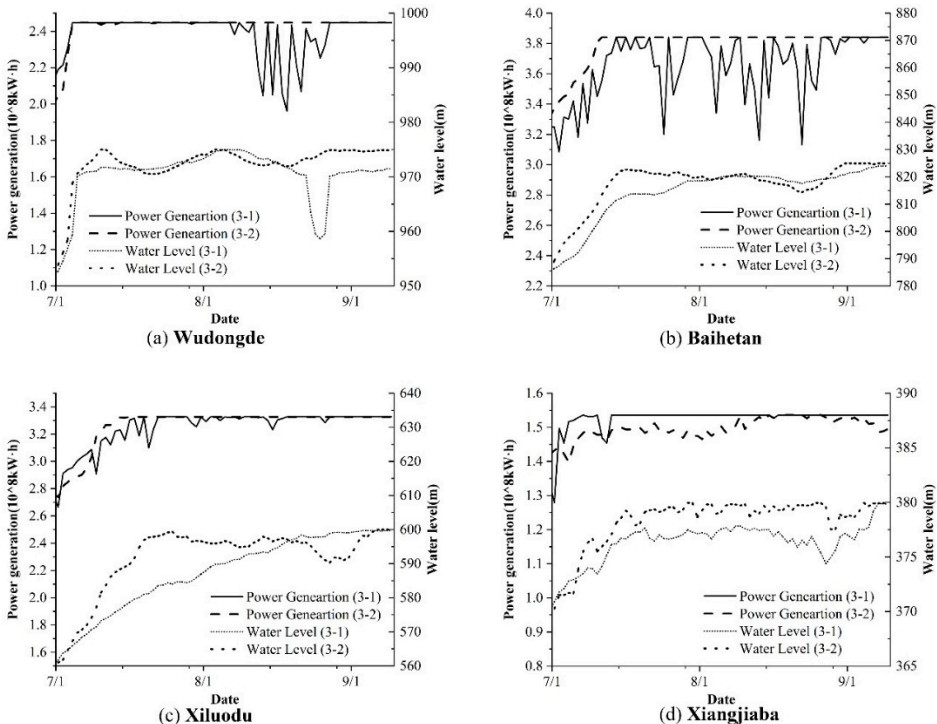

**Figure 9.** Water Level and power release trajectories of cascade reservoirs.

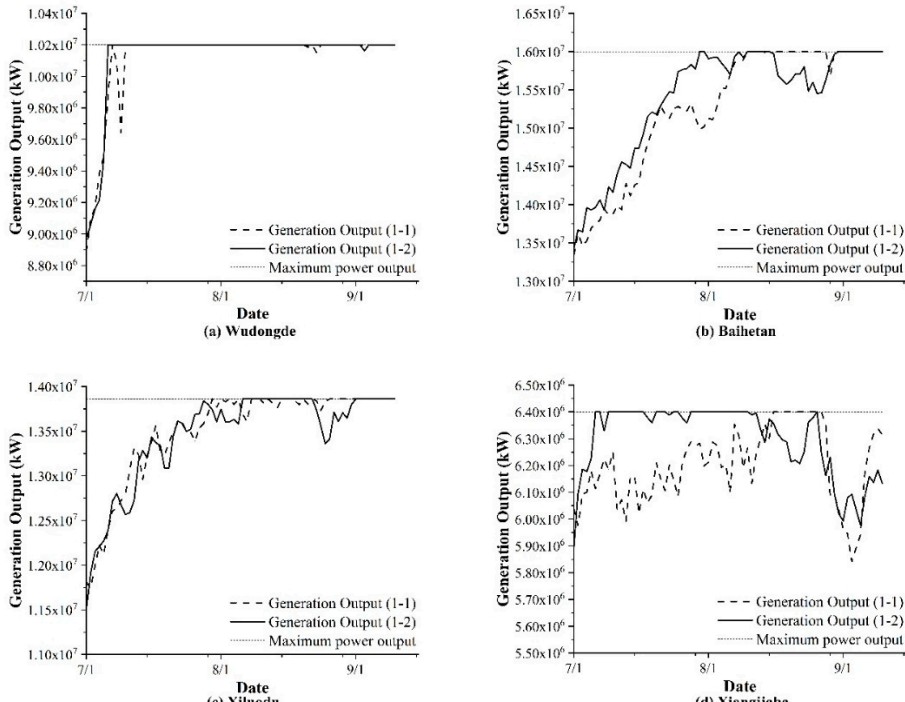

**Figure 10.** Generation output trajectories of cascade reservoirs.

Comparing the typical annual flood process and the amplified flood process in the same scenario, the internal competition and the rules of variation are similar in the flood season, but the competition intensity of the two objectives in the typical period of the early flood season is obviously stronger

than that of the amplified flood process. The reason is that in the competitive strength coefficient, $(\Delta t/(\Delta q\_4))^{\hat{}}2$ is the reciprocal square of variation of flow volume, which changes with time, so the value is much smaller in the amplified process than that in the typical process. Meanwhile, $\Delta E/\Delta t$ and $(\Delta q\_4^{\hat{}}2)/\Delta t$ make no significant difference, causing that competitive strength coefficient to be larger in the early stage of the typical process than that of the amplified process. From the perspective of the dispatching process, the large amount of water can "weaken" the demand for water and water heads, which makes the competition intensity weaken between power generation and flood control.

## 6. Conclusions

This paper mainly proposes a method for judging the competitive relationship in multi-objective optimization scheduling, establishes the mapping relationship between the varying trend of the objective function and the competition relationship, and proposes the derivation formula of the competitive strength coefficient, which avoids the previous study of the competition only from Pareto solution sets. Taking the cascade reservoirs in the lower reaches of the Jinsha River as an example, the NSGA-II intelligent algorithm was used to solve the flood control-power generation optimal dispatch model of three different inflow conditions each in typical years, and their flood peak discharge combined with the amplified flooding process (six types in total). The competitive relationship between flood control and power generation in different stages was analyzed, and the reasons for competition were analyzed through the dispatch hydrograph. The main conclusions are as follows:

(1) The power generation and flood control targets have a strong competitive relationship in the early stage of the flood season. The main reason is that the reservoir system needs to raise the water level, but it also needs to be properly discharged due to flood control. The essence of the competition between the two objectives is the distribution of water storage. The competition in the middle and late flood season depends on whether the water supply is sufficient or not: The sufficient water supply keeps the turbine unit full and weakens the competition.

(2) During the process of flood peak at the beginning of the flood season, the competition intensity in the early stage of the flood season is stronger than that in the later stage because of the large variable range of reservoir storage in the early period.

(3) The difference in flood process has little effect on the general trend of competition, but it has an impact on the duration and intensity of each stage. During the flood process of the typical year in 1974 and its amplified flood process, the strong competition relationship between the two objectives of flood control and power generation is much longer than other inflow conditions. At the end of the period, due to the arrival of flood peak, there is a strong competition period of about 10 days, and the competition intensity is stronger than the beginning of the period.

(4) In the same inflow process, the greater the flood volume, the weaker the competitive relationship in the early period. The large flow and high water head caused by large water volume weaken the competitive relationship between power generation and flood control.

Although this paper proposes a new method for discriminating the competitive relationship, there are still simplifications and neglects in many places, such as the lack of physical meaning for the composition of the competitive strength coefficient, and the lack of quantitative descriptions of the impact of the allocation of water storage on the competitive relationship. The above are all worthwhile to study further in the future.

**Author Contributions:** Conceptualization, H.Y.; Formal analysis, H.Y.; Funding acquisition, Z.D. and D.L.; Methodology, H.Y.; Project administration, Z.D. and W.J.; Software, H.Y., X.N. and M.C.; Supervision, H.Y., Z.D., W.J. and D.L.; Validation, C.Z.; Visualization, M.C.; Writing—original draft, H.Y.; Writing—review & editing, H.Y. and X.N.

**Funding:** This research was funded by the National Key Research and Development Project of China, grant number 2016YFC0402209.

**Conflicts of Interest:** The authors declare no conflicts of interest.

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
