# Peer review of "Competitive Relationship between Flood Control and Power Generation with Flood Season Division: A Case Study in Downstream Jinsha River Cascade Reservoirs"

_water, doi:10.3390/w11112401_

Round 1
Reviewer 1 Report
Major Comments
The following points should be looked at and addressed before the paper is publiched:
As noted in line 198 there are six scenarios used in the analysis. For each of the 1974, 1981 and 1985 years there are two scenarios, one called measured flood process and the other called amplified flood process. There is no explanation of what changes were made to the measured flows to produce the amplified flood process scenario. This is a major feature in the paper and a paragraph should be added to describe how the amplified flood process was created.It appears that in Table 1 {line 102}in the second third fourth and fifth column the headings of upper and lower bounds are reversed i.e. the second and fourth columns should be lower bounds and the third and fifth columns should be upper bounds.
For Equation (1) {line 109} there is no explanation, or any reference given, for the use of the sum of squares of discharge as a measure of flood damage. There should be an explanation with reference to published studies that develop and use this measure of flood damage.
In the reference list at the end of the paper et al. is used – in most standards for referencing all authors are named in the list of referenced sources and et al. is used only in the body of the paper when work is being referenced.
Editorial Comments
Line 34 reservoirs to prevent floods reduce flood risks.
Line 36 exploiting and utilizing reservoir operations to reduce flood damage flood resources improves.
Line 60 objectives is still has been given limited attention and its rules and variations are unknown.
Line 238 In Section 4, including the y axis label in Figure 5 the term Flood Risk is used when the term should be Flood Damage.
Reviewer 2 Report
Main comments
The manuscript submitted for review presents the problem of the competitive relationship between flood control and power generation. A case study (?) of Downstream Jinsha River Cascade Reservoirs was presented. In addition, models are used whose description is fragmentary. The authors use sorting genetic algorithm (NSGA-II) - this requires a more detailed description. Looking through the archive, you can also find the article recently published in Water by the co-authors dealing a similar problem on the example of the same case study (https://www.mdpi.com/2073-4441/11/4/849). The reviewer would like the authors to respond to this comment and indicate the difference between the used approach. Whether the use of a similar methodology and the use of the same case study, setting only different objective functions, has a scientific connotation?
The manuscript has many minor editing errors (list found below). It is particularly puzzling that errors are in references to literature. Please check that all abbreviations are defined when you first enter them.
Specific comments
Line 46 - Invalid reference to [9]
Line 66 - Invalid reference to [16]
Line 84 - in the title of chapter 2 the authors announce the presentation of materials and methods - there is only a description of the study area
Line 135 (equation 7) - instead of a nabla operator (Ñ) there should be a differential operator D
Line 193 - Invalid reference to [29]
Line 237 - illegible descriptions in table 4
Line 241, 273, 305, 317 - poor quality of drawings (figure 5,6,7,8)
Line 398 - Journal of Biological Chemistry???? - Journal of Hydraulic Engineering
Line 436 – Deb, K.
Round 2
Reviewer 2 Report
The authors have included all the comments in the current version of the manuscript. If the editor allows the article to be published, then I agree to the publication after checking all editorial defects